# Searching for Silphium: An Updated Review

**Lisa Briggs [1,*] and Jens Jakobsson [2,*]**

1   The British Museum, Great Russell St, London WC1B 3DG, UK
2   Independent Researcher, SE-22456 Lund, Sweden
*   Correspondence: ebriggs@britishmuseum.org (L.B.); jens.jakobsson@skola.malmo.se (J.J.);
    Tel.: +44-(0)-7940-418-110 (L.B.)

**Abstract:** From luxury spice to medical cure-all, silphium was a product coveted throughout the ancient world and occupied an essential place in the export economy of ancient Cyrene. The mysterious extinction of the silphium plant in the 1st century CE leaves us with little evidence as to the exact nature of this important agricultural product. In this paper, an historical background on the kingdom of Cyrene is provided, evidence for the nature of the silphium plant is reviewed, how and why it was consumed and traded is discussed. Possible causes of extinction are considered in the context of plant genetics, biometrics, and soil geochemistry. Next, we demonstrate how modern medical studies conducted on possible living relatives can inform us about claims made by ancient authors as to the medical uses of the silphium plant, including its use as a contraceptive and abortifacient. Finally, methods for recovering silphium are explored. We show how underwater archaeology and the search for ancient shipwrecks off the northern coast of Libya may offer our best chance for the recovery of botanical remains of ancient silphium, and how ancient DNA may be able to establish the genetic makeup of this elusive plant.

**Keywords:** silphium; silphion; archaeobotany; shipwrecks; Cyrene; contraception; abortifacient

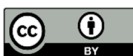

## 1. Introduction

Silphium was a highly coveted agricultural product that was used as a spice and as a cure for a variety of medical ailments. Widely discussed by ancient authors including Herodotus, Strabo, Pliny, Theophrastus, and Dioscorides, it was said to cure such a wide variety of medical ailments that it was extolled as a panacea [1]. Many plants found in and around the Mediterranean were transplanted or propagated by humans, resulting in a wide dispersal by the Late Bronze Age. Pomegranate is one such example, with its origins in the East and the Levant, and archaeological evidence demonstrating the progression westward of pomegranate cultivation and exchange throughout the Bronze Age [2], while the propagation of grapevines and olive trees in marginal environments has been argued to be a prime factor in the success of Early Bronze Age communities [3]. Garlic was so popular with the Roman army it has been said that the path of Roman legions around Europe could be correlated with a range map of garlic [4]. Silphium, conversely, was not cultivated, and instead grew wild on a narrow strip of land between the hilly hinterland and the Mediterranean Sea on the northern coast of what is now Libya, in the vicinity of the city of Cyrene. The importance of this export to the prosperity of Cyrene is evidenced by the imagery of silphium on the silver coins minted there, which endured as a motif on the city's currency for hundreds of years [5]. The silphium plant's resistance to transplanting and clonal propagation may have contributed to its status as a luxury food item; with a limited supply growing wild in a finite geographical area, it was not possible to mass-produce silphium in the ancient world, meaning the demand far outstripped the naturally available supply.

The end of wild silphium may be the first recorded case of a plant extinction in world history (4), and its causes remain unknown. The unexplained extinction of silphium in the

1st century CE has been cited as a reason for the economic decline of Cyrene, just as the early export of this plant was a central cause for the city's success in the 7th and 6th centuries BCE [6]. Given the exceptional economic importance of silphium, the myriad medical properties attributed to it, its status as the first-ever recorded species extinction, and new insights into plant genetics and biodiversity, a review of the evidence for silphium is necessary.

In this paper, we provide an updated review of the evidence for the silphium trade and the historical background of the city of Cyrene, which enjoyed significant economic benefits from the export of this elusive plant. We then consider possible causes for the plant's extinction, including issues of climate change, over-harvesting, genetic recombination, and soil geochemistry. Next, we consider the multitude of medical benefits of silphium extolled by the ancient authors, notably its use as a contraceptive, abortifacient, and possibly as an aphrodisiac. These claims are reviewed in light of modern medical studies conducted on what may be related species, and whether or not the results from these studies may back up these ancient claims. To conclusively determine the nature and genetic makeup of silphium, we require preserved remains of the plant itself. The recovery of preserved silphium remains will most likely come from the field of underwater archaeology. The identification and excavation of ancient shipwrecks departing from Apollonia, or submerged harbour areas of Apollonia, may provide our best chance for the recovery of archaeological remains of silphium, a species that Pliny called a 'very remarkable plant', which was 'sold at the same rate as silver' [7].

## 2. Historical Background

### 2.1. Cyrene and Its Agricultural Hinterland

The region of Cyrenaica in today's eastern Libya is characterised by discrete pockets of arable land, divided by inhospitable desert, which led to multiple waves of settlement, colonisation, and cultural developments. Evidence of pottery production and animal husbandry in the Neolithic period show there were early communities in the area, and it has been noted that the local culture of Cyrenaica differed markedly from that in contemporary Tunisia [8]. Herodotus reported that the first Greek settlers in the area consisted of Dorian Greeks from the island of Thera in the 7th century BCE [9]. According to Herodotus, the traditional founder of this colony was Battus I, who was allegedly told by an oracle to settle in Libya. There, he founded the city of Cyrene (modern-day Shahat), which became the centre of a small state, where his dynasty, the Battiads, ruled from c. 630–440 BCE. The local population of Libyans sought to limit the expansion of the Battiads and requested protection from the Egyptian king Apries, but the army that he sent was swiftly defeated [5]. Alliances shifted soon after, when the Battiads were allied to the Egyptian Pharaohs of the 26th dynasty, such as Amasis II, until the Achaemenids conquered Egypt in 525 BCE. After that, the last rulers became Persian vassals. In c. 440 BCE, the Battiad Dynasty came to an end when Archiselaos IV was killed at Euesperides [10], at which point the monarchy was abolished and Cyrene became a polis state. Following Alexander the Great's conquest of Egypt in c. 330 BCE, the city came under Ptolemaic suzerainty [11]. When Ptolemaic power waned, the last ruler, Ptolemy Apion—a natural son of Ptolemy VIII Physcon—bequeathed the province to the Romans, meaning soon after he died in 96 BCE, Cyrenaica became a Roman province. This province was also referred to as *pentapolis* after its five main cities: Cyrene, the port of Apollonia, Eusperides, Arsinoe, and Barca, also known as Barce [12]. The locations of these cities are shown in Figure 1.

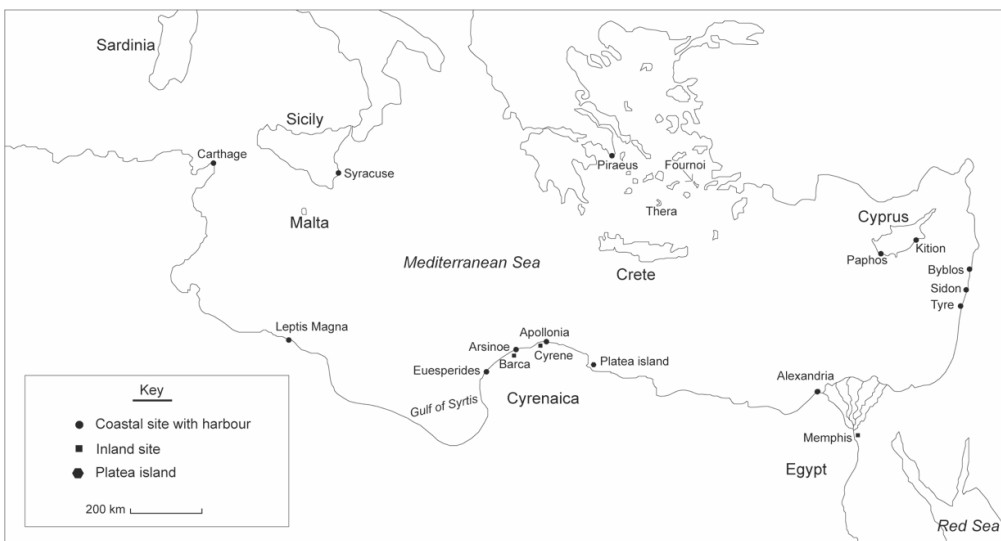

**Figure 1.** Map of Cyrenaica and relevant ports and cities mentioned in this text.

Cyrene and its agricultural hinterland had a favourable climate for agriculture, producing a number of products for export to the wider Greco-Roman world including wheat, a variety of legumes [13], barley, rice, onions, garlic, cumin, saffron, white violet, roses, cucumber, tree moss, grapes, and edible fungus [14,15], while animal-derived exports included livestock such as horses, cattle, sheep, goats, mules, and camels, as well as animal products including hides, ostrich plumes, and murex shells [14,15]. There is ample archaeological evidence for olive oil production in the region, including the remains of hundreds of olive oil presses, which further attest to the agricultural fecundity of this region [10]. Unlike many of the agricultural products listed above, silphium grew wild and uncultivated. Despite this impressive list of exports that fuelled the vibrant economy of Cyrene, it was silphium that was selected to appear on the silver coins of this kingdom and remained employed as the principal heraldic symbol of Cyrene on their silver coins for over three centuries [5].

Theophrastus describes how silphium grew throughout Cyrenaica in several sections of his widely quoted work, *Historia Plantarum*, but it grew most profusely in an area dubbed *Silphiofera*, located near the Gulf of Syrtis (Figure 1), where the plateaus rise off the Mediterranean coast and receive greater levels of rainfall than in the deserts to the south [4]. These hilly and forested meadows allowed the wild plant to thrive, and likely also served as optimal areas for sheep and goat grazing. Both sheep and goats were among the many exports of Cyrene [14], and Theophrastus described how the flesh of these animals tasted more delicious if they were allowed to graze on silphium plants [16]. However, the population of silphium dwindled at least from the Roman period, and it was regarded as extinct in the late 1st century CE by Pliny, who described how Emperor Nero was given a single stalk of the plant, as a curiosity [7].

However, later scholars such as Galen and Aelian kept referring to 'the juice of Cyrene', and it appeared as an ingredient in Apicius' cookbook [17]. It is uncertain whether these scholars had access to late specimens of actual silphium plants or silphium resin, or if they used some substitute plants such as asafoetida. Long after the purported date for the extinction of silphium in the 1st century CE, this plant and its uses were described in European and Arab texts, both culinary and medical [17], by authors who are unlikely to have ever encountered genuine silphium from Cyrenaica. For example, early 5th century CE sophist philosopher, Bishop of the *Pentapolis*, and prolific letter-writer, Synesius of Cyrene, described a shipment of goods including saffron, olives, ostriches, and silphium that failed to make it to its destination due to tribal raids [10]. This begs the question as to whether silphium truly did become extinct in the 1st century CE.

## 2.2. Excavations in Cyrenaica

Early excavations at Cyrene revealed a wealth of artistic evidence for silphium, principally in the form of ceramic figurines of women holding silphium plants, of which hundreds were recovered in the first expeditions in the 1910s [18], and silver coins that featured the silphium plant more than any other motif [5]. In addition, stamped ceramic bowls with a shallow foot bore a characteristic silphium stamp, often impressed multiple times around the centre of the container [18]. This assignation was challenged recently, however, when a review of the original site reports suggested that these ceramic vessels were, in fact, black glaze kylikes with palmette decorations, rather than depicting silphium in the ceramic decoration [19]. Sadly, these early excavation reports focused almost exclusively on objects of artistic value, with no mention of any charred, carbonised, or desiccated organic remains in the form of seeds or other plant material. Any direct evidence for the organic remains of silphium was likely lost on the spoil heap of time. More recent excavations at the Sanctuary of Demeter in Cyrene, conducted in the 1970s, did not record any organic remains recovered by flotation or sieving, but they did attest to the great wealth and culture of Cyrene based on the number of marble statues, quantity of dressed stone, and the impressive size of the architectural remains recovered [20].

While the city of Cyrene has obvious and principal importance in the region as the first city founded by Greek colonists (according to Herodotus, among others), and from which the region of Cyrenaica derived its name, there are four other important trading centres in the area. Ancient Euesperides, which is modern-day Benghazi, has been the site of relatively recent excavations in the region. Identified in 1948 by archaeologists from the Ashmolean Museum, and excavated throughout the 1950s, 1960s, and 1990s by archaeologists from the University of Oxford, this site is of exceptional importance for understanding the early history of Cyrenaica [21].

The earliest ceramics found at the site are of an ancient Corinthian style and date back to circa 620–600 BCE [21]. Some amphorae have been uncovered at the site with figurative stamps that may resemble silphium, and therefore, were probably intended to contain the precious silphium resin for which this region was renowned [21]. While silphium grew nearby to Euesperides and undoubtedly formed a part of the ancient city's export economy, abundant archaeological evidence points to murex collection and processing being of principal importance to the ancient economy of Euesperides [21]. Botanical remains recovered through flotation at the site of Euesperides have been analysed, with barley, grape, and fig remains dominating the assemblage [22].

The port of Apollonia served as the harbour town for the nearby city of Cyrene [23], and it is from this location that we would expect exports of silphium to have emanated. Excavations at Apollonia conducted by the University of Michigan revealed bathhouses, towers, and city walls, as well as numerous coins, pieces of glass, and ceramics [24]. Unfortunately, no mention was made of any botanical remains recovered, nor of sieving or flotation performed on any of the spoil accumulated during these excavations that might have yielded botanical remains of silphium. However, the numerous ceramics recovered at Apollonia may have once contained silphium gum or resin. Analysis by gas chromatography-mass spectrometry (GCMS) could help shed light on what these vessels contained and if it is possible to find molecular evidence for silphium in the ceramics discovered at this important harbour city. Fortunately, seismic activity has caused sections of the ancient harbour infrastructure to fall into the sea [23].

Underwater investigations of the harbour at Apollonia were conducted in 1958 and 1959 by a team from Cambridge led by Nicholas Flemming. The team conducted a thorough survey of both the submerged land surface and the harbour infrastructure, where evidence was found of ship slipways, submerged quay structures, stone anchors, and both an inner and outer harbour [25]. After a review of Flemming's site plans and interpretation of the harbour of Apollonia, we surmise that the section identified as the 'Inner Harbour' is the most likely to contain waterlogged remains of cargo items. The fact that significant sections of the harbour infrastructure are now underwater may increase our chances of

finding silphium remains as botanical remains are more likely to survive in underwater archaeological contexts [26–28].

### 2.3. Sources of Information on Silphium

Our primary sources of information on the morphological attributes of silphium derive from numismatics [14], and to a lesser extent, terracotta figurines from Cyrenaica that typically depict a female figure holding a silphium stalk. The silver coins of Cyrenaica were thoroughly catalogued, with the earliest dating from c. 560 BCE [5]. Silphium, be it the entire plant, a fruit, flower, leaf, or seed, almost invariably appears on the obverse, and sometimes is duplicated on the reverse of these coins [5]. The fruit was the first element of the plant depicted on the earliest coins, replaced by other parts of the plant in later periods [5]. The silphium fruit was depicted as 'heart' shaped (Figure 2).

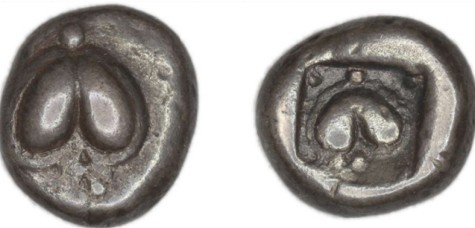

**Figure 2.** Silver coin of Cyrene, c. 525–480 BCE. Museum number RPK,p97E.1.Car. Copyright: Trustees of the British Museum.

In Figure 3a, a relatively early rendering of the entire silphium plant is shown. This may be a more naturalistic depiction as this version shows the stalk alternately tending towards the left and then the right, while the leaves are depicted opposite the umbels. An umbel is defined as a flower cluster where stalks of equal length emanate from a common centre, as is characteristic of the parsley family of plants. Later depictions of the silphium plant are markedly different from this early rendering.

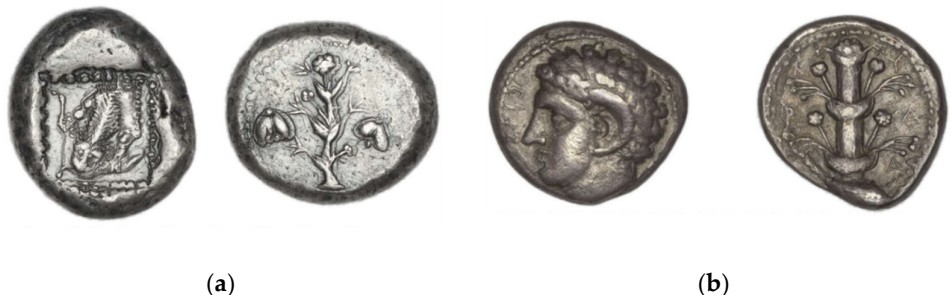

(**a**)                                                 (**b**)

**Figure 3.** Silver coins from Cyrene showing how the depiction of the silphium plant changed over time. (**a**) Silver coin of Cyrene, c. 525–480 BCE depicting silphium plant (obverse) and lion devouring prey (reverse). Museum number 1971,0513.12. Copyright: Trustees of the British Museum; (**b**) Silver coin minted at Cyrene depicting the head of Ammon (obverse) and silphium plant (reverse), c. 435–375 BCE. Museum number 1928,0120.98. Copyright: Trustees of the British Museum.

As shown in Figure 3b, depictions of the silphium plant became more stylised in later periods. This coin, minted at Cyrene between 435 and 375 BCE, shows a very thick stalk that terminates in an umbel with poorly rendered flowers. The appearance of the plant is altered when compared with the earlier coin shown in Figure 3a, which we argue is due

to a gradual shift towards a less realistic and more stylised artistic rendering of the plant. This variety in renderings may simply be a function of phases in artistic styles, which may have previously favoured more realistic depictions of plants, animals, and people, while in subsequent phases, idealised renderings became more commonplace. As this later style of depicting the silphium plant is more widespread, with greater numbers of coins surviving that show this later style, we argue that this may have altered the perception of the gross anatomical morphology of the silphium plant in the minds of those seeking living relatives of the silphium plant in modern times. It is our opinion that the earlier version shown in Figure 3a more accurately reflects the shape of the ancient silphium plant due to this version showing greater similarity to modern *Ferula* species (Figure 4).

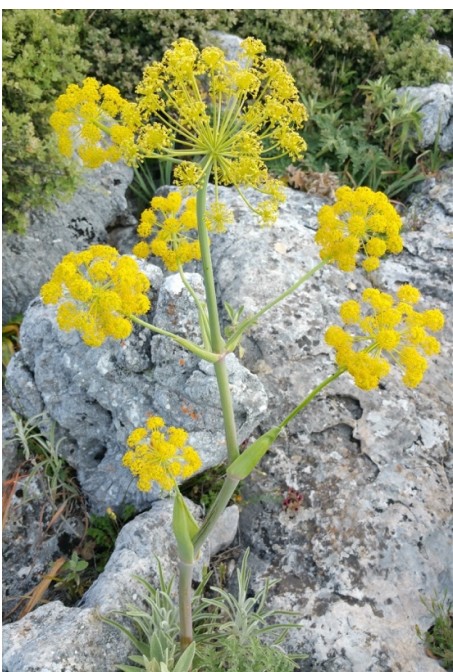

**Figure 4.** Modern *Ferula tingitana* plant. Wiki Commons media, https://commons.wikimedia.org/wiki/File:Ferula_tingitana.jpg Reuben0568, accessed 14 March 2022.

While images on silver coins may not give an accurate sense of scale, terracotta figurines may indicate the size of a typical silphium stalk. Hundreds of terracotta figurines of a woman holding a stalk of silphium were recovered during the excavations of Cyrene undertaken in the 1910s [18]. Such figurines may offer a sense of scale for the silphium plant. (Figure 5). This figurine depicts the city of Cyrene as a goddess, holding the plant with which her city was so intimately associated.

If this depiction of Cyrene is meant to show her in human form at a realistic scale, we expect that the stalk of the plant could be easily grasped and that the leaves and umbels were not overly large. However, as this may be a stylised depiction, any attempts to determine the scale are merely speculative.

Another source of information on the silphium plant comes from ancient authors who either described it outright or made reference to it in plays or literary works. Asciutti (2004) provided a thorough survey of ancient authors on silphium, from both the Greek and Latin literary corpora, so this is not repeated here. Below is a selection of the most relevant references to silphium by ancient authors, while a table of additional references by ancient authors is provided in Appendix A, Table A1.

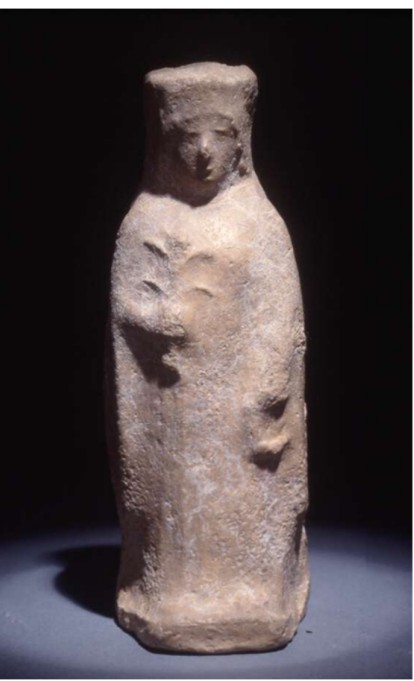

**Figure 5.** Mould-made terracotta figurine depicting Cyrene as a woman holding a stalk of silphium in one hand and a sickle in the other. Museum number 1879,0405.2. Height 14.5 cm. Copyright: Trustees of the British Museum.

Herodotus described Cyrenaica as being defined by its silphium in the passage that stated, '*Here the country of silphium begins, which reaches from the island of Platea to the entrance of Syrtis*' [9]. This passage suggests an exclusive connection between Cyrenaica and silphium, not shared by other regions in North Africa, or indeed other parts of the Greek world.

The playwright Aristophanes made several references to silphium in his plays, including works such as *Birds*, *Knights*, and *Plutus* [29]. In his play *Birds*, the character Peisetairus prepares a sauce for roasted poultry, seemingly ignoring three Gods who have just entered the scene, Poseidon, Heracles, and Triballus. He calls out to his slaves, 'Hand me the grater, the silphium, and the cheese, now poke the fire!' Poseidon interjects with, 'A greeting to you, my man. We are a threesome of gods.' Peisetairus, appearing disinterested, replies 'I'm grating silphium.' This passage would have been humorous to Athenian audiences for two reasons: first, ignoring a threesome of Gods to make a tasty sauce, was, and is, funny, but the second part illuminates another humorous aspect, which is the fact that silphium was expensive and imported, and even the presence of the Gods was not sufficient cause for Peisetairus to stop grating his precious silphium. Finally, this illuminates a physical property of silphium. It is likely that the resin of this plant was transported in lumps that were morphologically similar to raw asafoetida resin today, or the Pistacia terebinthus resin and mastic resin transported widely in antiquity. To powder the silphium resin, a grater was necessary. A passage in another of his plays, *Knights*, lampoons the leader Cleon when a sausage-seller describes the time that Cleon made silphium 'so cheap' (line 895) that it caused all of Athens to be uncomfortably flatulent [29].

Theophrastus provided detailed information on how the gum or resin of the plant was obtained by cutting part of the root and allowing the sap to collect until it hardened into a resinous lump: '*The juice of silphium is pungent like the plant itself; for what is called the 'juice' of silphium is a gum*' [16]. He went on to describe how the 'juice of silphium' was obtained: '*In those plants whose stem and root are both cut the stem is cut first, as also with silphium; and the juices so obtained are called respectively stalk-juice and root-juice, of which the latter is the better, for it is clear transparent and less liquid. The stalk-juice is more liquid, and for*

*this reason they sprinkle meal over it to make it clot. The Libyans know the season for cutting, for it is they that gather the silphium'* [16]. Of special importance is how Theophrastus described the method of conveying silphium abroad. As all ancient authors seem to agree, true silphium only grew in Cyrenaica. Therefore, any silphium consumed abroad was exported from this region. Theophrastus described how it was exported: '*When they are conveying it to Peiraeus, they deal with it thus:—having put it in vessels and mixed meal with it, they shake it for a considerable time, and from this process it gets its colour, and this treatment makes it thenceforward keep without decaying'* [16]. Based on this description, we expect silphium gum or resin to have been transported to Piraeus in amphorae onboard ships, in a similar way to how *Pistacia terebinthus* resin [30], pine resin, and pine pitch were transported [31].

Herodotus, Aristophanes, and Theophrastus were writing at a time when silphium was still actively traded and consumed around the Mediterranean. Pliny the Elder also provided an extensive description of silphium in his *Natural History*, but this was written at a time when the plant was thought to have already gone extinct: '*…laserpitium claims our notice, a very re- markable plant, known to the Greeks by the name of "silphion," and originally a native of the province of Cyrenaica. The juice of this plant is called "laser," and it is greatly in vogue for medicinal as well as other purposes, being sold at the same rate as silver. For these many years past, however, it has not been found in Cyrenaica, as the farmers of the revenue who hold the lands there on lease, have a notion that it is more profitable to depasture flocks of sheep upon them. Within the memory of the present generation, a single stalk is all that has ever been found there, and that was sent as a curiosity to the Emperor Nero. For this long time past, there has been no other laser imported into this country, but that produced in either Persis, Media, or Armenia, where it grows in considerable abundance, though much inferior'* [7]. This passage suggests that similar plants and products replaced silphium, which likely included the more pungent asafoetida, still regularly employed in the cuisine of the Indian subcontinent today [32].

## 3. Medical Effects—Real or Imaginary?

Classical sources describe a wide variety of medicinal uses for silphium [14], to the extent that silphium was described as a panacea, or cure-all, by the Roman author Lucan [1]. Indeed, ancient sources describe silphium as having therapeutic properties that can mitigate the adverse effects of virtually every type of ailment, from skin problems to respiratory illness, dog bites to epilepsy. Curiously, it was effective for both hair removal and as a hair restorative [14]. Interestingly, some ancient sources describe effects of silphium that are conspicuously similar to the effects of asafoetida, which was described even in ancient times as a replacement for silphium if the former was unavailable [7]. As discussed above, Aristophanes poked fun at Cleon by claiming that when the Athenian tyrant lowered the price of silphium, it caused the entire population of Athens to become more flatulent. The expelling of 'wind from the stomach' was a traditional use of asafoetida that carries on in the Indian subcontinent today [32].

While a wide range of medical uses for silphium was listed by ancient authors, it was said to cure a conspicuous number of reproductive issues, including acting as an abortifacient, contraceptive, for menstrual problems, and as a purgative to expel the placenta or a stillborn foetus [14]. Here, we explore how and why silphium came to be seen as an aphrodisiac, contraceptive, and abortifacient, and whether or not there may be any medical basis for these suppositions.

### 3.1. Aphrodisiac

Modern scholars tend to associate silphium with carnal pleasures in the ancient world and discuss how it may have been used as an aphrodisiac, contraceptive, or abortifacient. However, an exhaustive look at references to silphium by ancient Greek and Latin authors by Asciutti (2004) revealed that there are, in fact, no extant references by classical authors that directly state this plant served as an aphrodisiac [1]. Some have speculated that the silphium plant had no inherent aphrodisiac properties but instead garnered this reputation due to the phallic and testicular shape of the plant itself [14].

The first scholarship that proposed silphium as an ancient aphrodisiac came in the 19th century from Ibn Sina, known as Avicenna, an author working out of Cairo who produced a volume in Arabic called 'A System of Medicine' [33]. Modern scholars have speculated that Avicenna referred to 'Eastern silphium', meaning asafoetida, and this misunderstanding is what propelled the Victorian era scholarship on the ancient silphium of Cyrene and its role as an aphrodisiac [14]. It seems likely that the association between silphium and aphrodisiacs came about due to its perceived status as a contraceptive. However, as will be explored below, related species of *Ferula* exhibit properties that alleviate erectile dysfunction, and if ancient silphium had similar properties then perhaps this is the source of its association with sexual activity.

### 3.2. Contraceptive or Abortifacient

Several ancient authors describe its uses as an oral contraceptive and abortifacient, including Dioscorides of Anazarba and Soranus of Ephesus [14], while Pliny the Elder describes the plant's ability to expel an unborn foetus: *'It originally came from Cyrenæ, as already stated: at the present day, it is mostly imported from Syria, the produce of which country, though better than that of Media, is inferior to the Parthian kind. As already observed, the silphium of Cyrenæ no longer exists. It is of considerable use in medicine, the leaves of it being employed to purge the uterus, and as an expellent of the dead fœtus; for which purposes a decoction of them is made in white aromatic wine, and taken in doses of one acetabulum, immediately after the bath'* [7].

Pausanias, a geographer writing in the 2nd century CE, made reference to the use of silphium in the distant past, in a passage that not only provides insight into this elusive plant but also illuminates attitudes surrounding female chastity. As marriage in the ancient Greek world primarily served to provide legitimate offspring, the value of a daughter could be diminished or destroyed by even the suggestion of illicit sexual encounters, whether those encounters were consensual or not [34]. In her work on the language of Pausanias and his use of several terms that can be translated as 'rape', Cundy (2021) examined the context for otherwise ambiguous passages from Pausanias in detail. In a passage that detailed a visit to a mortal man, Phormion, by the semi-divine Dioscuri twins, Pausanias described the visitors: '*They said that they had come from Cyrene and asked to lodge with him, requesting to have the chamber which had pleased them most when they dwelt among men. He replied that they might lodge in any other part of the house they wished, but that they could not have the chamber. For it so happened that his maiden daughter was living in it. By the next day this maiden and all her girlish apparel had disappeared, and in the room were found images of the Dioscuri, a table, and silphium upon it'* [35]. Asciutti (2004) posited that the silphium in this passage simply represented Cyrene, the place of origin for the Dioscuri men. However, there is another possibility. This could be an oblique reference to its properties as a contraceptive or abortifacient, in which case preventative measures were being taken to ensure the maiden daughter did not fall pregnant. While this specific passage is not included in the study by Cundy (2021), it appears to be relevant to investigations into how silphium was used, and perceived, in the ancient world.

There is medical evidence to suggest that giant fennel and its relatives offer contraceptive properties. Several modern studies concluded that three species of *Ferula* (*Ferula assafoetida* L., *Ferula orientalis* L., and *Ferula jaeschkeana*) exhibit anti-fertility properties in rodents [36–38], while the resin of the asafoetida plant has been reported in human tests as acting as a contraceptive and abortifacient [39]. If this is the case, it strengthens our evidence for ancient silphium sharing similar properties. As discussed above, numerous ancient sources described how asafoetida could be substituted for silphium. This further suggests that asafoetida and silphium shared similar properties in taste and smell, and perhaps also shared contraceptive and anti-fertility properties. In a recent study, 31 active compounds were detected in what may be a related species in Turkey, *F. drudeana*, and their associated biological activities were investigated [40]. These included compounds with anti-inflammatory properties, antibacterial properties, and notably, the partial

elimination of erectile dysfunction [40]. These modern medical studies demonstrated the range of medical uses that can be attributed to *Ferula* species today, and likely attest that ancient silphium, if indeed a member of the *Ferula* plant group, harboured similar properties.

## 4. Causes of Extinction

Pliny the Elder stated that silphium was extinct by the time of writing his *Natural Histories* [7], however, later authors including Synesius of Cyrene, writing in the 4th century CE, referred to shipments of silphium in his lifetime [10]. While it is not clear whether or not silphium truly went extinct at this time, if it did, it would constitute the earliest recorded special extinction in world history [4]. In this section, we investigate possible causes of silphium's extinction.

### 4.1. Climate Change

The disappearance of silphium roughly corresponds with the period called the 'Roman Warm Period', c. 250 BCE–400 CE [41]. Strabo referred to the plains of silphium as being arid and sandy; the herb was apparently well-adapted to this, but an increase in temperature might have led to increased evaporation that made the conditions less favourable [42]. Evolutionarily distinct species of plants are at a greater risk of extinction in general, as are species of plants that occupy lower altitudes [43]. Since both were likely true for the silphium plant, it may have been exceptionally prone to the negative effects of climate change. Recent work that models future extinction events has shown that plant extinction due to climate change may be more likely than animal extinction, however, plant extinction is more likely to have a cascade effect of coextinctions, which would include dependent animal species [44]. If this 'Roman Warm Period' raised the local temperature by even a few degrees, it may have made the narrow area where silphium grew inhospitable for this delicate plant.

### 4.2. Overharvesting

Given the concentration of silphium in a small region, the possibility of overharvesting may well have contributed to its disappearance. Sheep were left to graze the leaves, and their hooves may have changed the composition of the topsoil. Theophrastus referred to how the roots should be cut, but not extensively: *'They have regulations, like those in use in mines, for cutting the root, in accordance with which they fix carefully the proper amount to be cut, having regard to previous cuttings and the supply of the plant'* [16]. Perhaps these regulations were originally put in place to discourage over-harvesting but were subsequently ignored. Dioscorides also discussed how the resin should be immediately treated with flour to preserve it so that it did not degenerate quickly [45,46]. It is possible that the extraction of the resin could be done by digging carefully around the plant, milking it, as it were, and then restoring the soil. That way, if the plants were perennial (as relatives like fennel, *Foeniculum vulgare*, still are), they could be drained many times. When the prices soared, croppers may have resorted to digging up the entire taproot for short-term gains.

### 4.3. Soil Characteristics

Little research has been conducted on the soils and sediments of Cyrenaica. However, it has been established that there are limited, but very fertile, pockets of arable land between the Gebel Akhdar and the coast, which were formed by rainfall collecting minerals and organic matter from the northern mountains and depositing these alluvial soils in the lowlands as these small rivers and wadis flowed towards the sea [10]. These fertile soils contrast with the more common soil type in the region, which consists of shallow, limestone-rich soils, poor in phosphorous [47].

It is possible that the soil in the region had special characteristics, which contributed to the specific taste and smell of silphium, as is the case for many exclusive strains of

grapes to this day. Erosion of the soil might then have changed the plant, making it less attractive for consumption, and perhaps leading it to no longer be identified as silphium proper. Palynological evidence has shown that the range of plants present in this part of Libya changed in the first century CE likely due to upland forest clearance to make more land for olive groves and cereal cultivation [10]. Upland forest clearance may have upset the delicate balance of soil characteristics and adversely affected the microbiota necessary for silphium cultivation.

Another possible cause of extinction for the silphium plant was a combination of changing temperature and soil characteristics. Temperature can have a drastic effect on soil geochemistry, which can lead to the extirpation of invertebrate communities within the soil [48]. Therefore, these two possible causes of extinction could have been linked. If the 'Roman Warm Period' caused a rise in temperature that adversely affected the delicate communities of bacteria, fungus, and invertebrates within the soil of Cyrenaica, this could have had a knock-on effect causing the soil to change in a way that meant it no longer supported the growth of silphium.

### 4.4. Polyploid or Hybrid Strains—Apomictic Production

It was suggested by Theophrastus that silphium was a newly discovered plant [16]. This remains unproven, and Gemmill analysed the origin of the name and found that the Greek term ΣΙΛΦΙΟΝ was older than the foundation of Cyrene [17], even though the name did not necessarily always refer to the same plant—as the ancient authors made clear, Syrian and Parthian silphium referred to other plants of inferior quality.

Nevertheless, new varieties of plants may appear within a short time due to genetic mechanisms that are well-known today. First, there is the process of polyploidy—fusion of gametes without the usual meiotic reduction of chromosomes [49]—which seems to have created several new culturally important species. In the *Brassica* genus, the species rapeseed (*B. napus*, 2n = 38) is thought to have emerged through allopolyploidy between *B.oleracea* (2n = 18) and *B.rapa* (2n = 20) [50]. Autopolyploidy—a change of the chromosomal number within the same plant—also gives rise to new varieties. Polyploids may exhibit limited fertility due to the mismatch of chromosomes. A search of the IPCN Chromosome Reports revealed that for c. 90 extant species of the *Ferula* genus, 2n was consistently 22. For the four *Thapsia* species, including *T.garganica*, it was similarly so, except for reports of 2n = 44 or even 66 for *T. villosa L*. Both numbers are multiples of 22, as would be common from autopolyploidi.

Even with the same number of chromosomes as its parents, silphium might have been a specific hybrid breed; such crosses may exhibit heterosis or hybrid vigour, a phenomenon well-known from many cultivated plants and a standard method to create new types. In this case, the result would supposedly have been increased production of molecules with a distinct taste and aroma.

While this remains unproven, genetic mechanisms might explain why silphium proper—i.e., the medical plant—was vulnerable. For this hybrid, outbreeding with other varieties of the plant might have caused the loss of some of the desirable traits, reverting the silphium to wild-type plants, which were not recognised as silphium by consumers—even though they were genetically the same species.

If this were true, it seems possible that asexual reproduction (sympodiums, apomictic reproduction) may have been crucial to preserve silphium proper. Hence, it would not have been easily possible to cultivate it through seeds in other areas, and damage to the original populations would have been difficult to overcome.

*4.5. Nomadic Raids*

Finally, Strabo claimed that nomad 'barbarians' had nearly destroyed the populations of silphium [42], but this seems difficult to prove. There might be some support for this claim from Roman author Lucius Ampelius [51] who stated that a son of Ptolemy VIII called 'The Cypriote'—both Ptolemy IX and his brother Ptolemy X spent time as rulers of Cyprus [52] and may fit this description—fought many wars against the Garamantes, an ancient people who controlled the arid regions of southern Libya [53]. These wars would have taken place soon after 100 BCE, and they may have included raids against the silphium fields, nominally within the realm of Ptolemy ('The Cypriote's' half-brother, Ptolemy Apion) until 95 BCE. Such disruptions to rural communities with expert knowledge of the silphium plants might have been just as devastating as actual destruction.

However, given Pliny told us that the Romans bought 30 pounds (c. 10 kg) of silphium in the 90s BCE, and as Julius Caesar is said to have taken the remarkable weight of 500 pounds (c. 490 kg) from the Roman treasury during the civil war in 49 BCE [7], it seems unlikely silphium was extinct after the raids conducted by the Garamantes soon after 100 BCE.

## 5. Searching for Silphium

*5.1. Alternate Species*

Several related species were also consumed during the classical period and remain so to this day, in particular, asafoetida, which remains a key ingredient in recipes of the Indian subcontinent [32]. It was the silphium of Cyrene, however, that commanded such a high price, compelling some ancient authors, including Synesius, to call it 'silphium of Battus' to differentiate true silphium from what he saw as inferior eastern substitutes [54].

Further contributing to the confusion surrounding the identity of ancient silphium is the fact that there is an entirely unrelated genus native to North America referred to as *Silphium*. These plants belong to the Compositae taxon and include rosinweed (*Silphium integrifolium*), which, interestingly, also produces an edible resin used by Native Americans for medicinal purposes.

*5.2. Candidate Plants—Hiding in Plain Sight?*

A phenomenon called 'plant blindness' has recently been explored as it relates to the realm of plant conservation. This phenomenon has been defined as 'a tendency among humans to neither notice nor value plants in the environment' [55]. This research has shown that humans are far more likely to both notice and recall animals than plants [55]. Perhaps the loss of silphium, or our inability to locate it in the modern world, could be down to plant blindness. If this is the case, silphium could very well be hiding in plain sight.

As discussed above, numismatic imagery in the form of thousands of silver coins from Cyrene [5] provide the best evidence for how the plant appeared, indicating that ancient silphium was morphologically dissimilar to the modern *Silphium* genus of North America. These extant images of the silphium plant on ancient coins and figurines bear a close morphological resemblance to the Apiaceae family of plants, also known as Umbelliferae. These include parsley, carrots, and several species of resin-producing flowering plants that are visually quite similar to depictions of silphium, such as species of giant fennel (*Ferula*), and so it is assumed that this genus is the closest living relative to the extinct silphium of Cyrene [12].

Candidates for silphium include various plants, some in the genus *Ferula*, as plants from this genus are frequently found in the region, which visually resemble silphium (Figure 4) and have pharmaceutically active substances. Gemmill (1966) suggested the following plants as either living relatives of Cyrenaican silphium or the plant itself:

- *Ferula tingitana*, or giant fennel (though it is not a proper fennel), which grows in today's Libya. It has potential medical effects including affecting the menstrual cycle [56].
- *Thapsia garganica*, which belongs to a different genus (*Thapsia* plants are sometimes referred to as 'deadly carrots') but has visual similarities with silphium and pronounced pharmaceutical properties, including as a cancer treatment. Söderling-Brynolf (1970) also suggested this plant, which was called 'drias' among peasants in the region of Cyrene and regarded as poisonous for livestock [57]. Interestingly, *Thapsia garganica* bears a further resemblance to ancient silphium in that it is extremely resistant to seedling propagation and must be micropropagated [58]. Given its distribution in the Gebel Al Akhdar area of Libya, this plant is an increasingly good candidate plant for ancient silphium. In addition, *Thapsia garganica* (Figure 6) produces fruits in a similar shape to ancient silphium fruits depicted on the earliest silver coins of the Battiad dynasty (Figure 2).

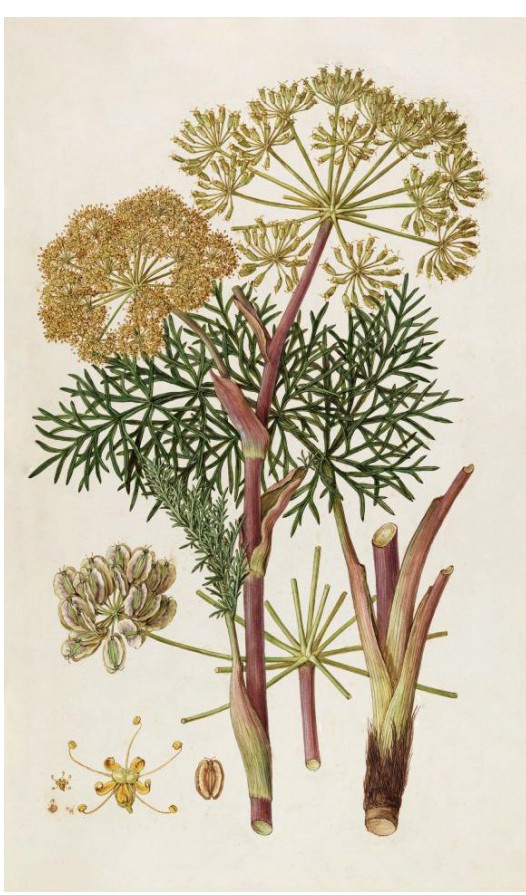

**Figure 6.** Drawing of *Thapsia garganica* by Baur. Creative Commons.

Depending on the genetic background for silphium, the relationship between these candidate plants and the elusive silphium of Cyrene could be either species in the same taxonomic family, hybrids, or the silphium itself, hiding in plain sight. Perhaps the silphium plant of Cyrenaica simply evolved to contain fewer of the properties that made it desirable and was increasingly replaced in use by cheaper substitutes from the East.

Recently, a new candidate species, *Ferula drudeana*, was proposed to be the actual silphium plant, still in existence, by a researcher in Anatolia [40]. This proposition was justified with the following: '*A rare and endemic species of Ferula growing near Central Anatolia closely resembles the description and numismatic figures of silphion*' [40]. The author went

on to describe how the oleo-gum-resin is similar to that described by ancient authors. While we certainly applaud efforts to recover silphium, these assertions must be thoroughly examined. The first assertion is somewhat problematic as *Ferula drudeana* is morphologically almost indistinguishable from other *Ferula* species and does not bear any greater resemblance to the crude depictions of silphium on silver coins than does any other candidate plant. The second assertion was that the gum-resin bears a close resemblance to silphium. Yet, just as the gross anatomical morphology of *Ferula drudeana* does not bear any greater similarity to silphium than other candidate species, it must be pointed out that virtually all *Ferula* species will produce a gum-resin exudate when either the stalk or the root is cut. As to whether or not the chemical properties of the gum produced by the various *Ferula* species are more or less similar to ancient silphium, it would require a full molecular study of each species to determine whether or not the gum-resin of *Ferula drudeana* contains more pharmacologically active substances than any of the other candidate species.

Finally, while it is certainly possible that the silphium of Cyrene did not go extinct and is hiding in plain sight, the candidate species we have proposed here occupy a similar geographical zone to the original plant. From Herodotus onwards, ancient authors made it clear that silphium only grew in a specific ecological niche between the island of Platea and the Gulf of Syrtis, and it seems unlikely that the silphium of Cyrene survived in Central Anatolia while dying out completely in North Africa. Yet, 'unlikely' is not 'impossible' and any research that considers the fate of silphium moves the field forward. Perhaps with further research, the assertions of Miski (2021) can be tested by comparing *Ferula drudeana* with the various other candidate species described above [40].

### 5.3. Underwater Archaeology and the Search for Silphium

The majority of silphium exported from Cyrenaica was by necessity taken on ships sailing maritime trade networks. Indeed, low-friction maritime transport swiftly began replacing the more arduous overland transport from the Bronze Age onwards [59]. As ancient ships conveyed silphium around the Mediterranean, some ships carrying silphium doubtless foundered before reaching their destination. If this is the case, there may be waterlogged remains of ancient silphium on the Mediterranean seafloor, which could provide direct evidence of the nature of this elusive plant. It is thus on the wrecks of ships that left Cyrenaica for distant harbours, or in the muddy bottoms of the harbours themselves, that we are most likely to recover organic remains of ancient silphium (Figure 7).

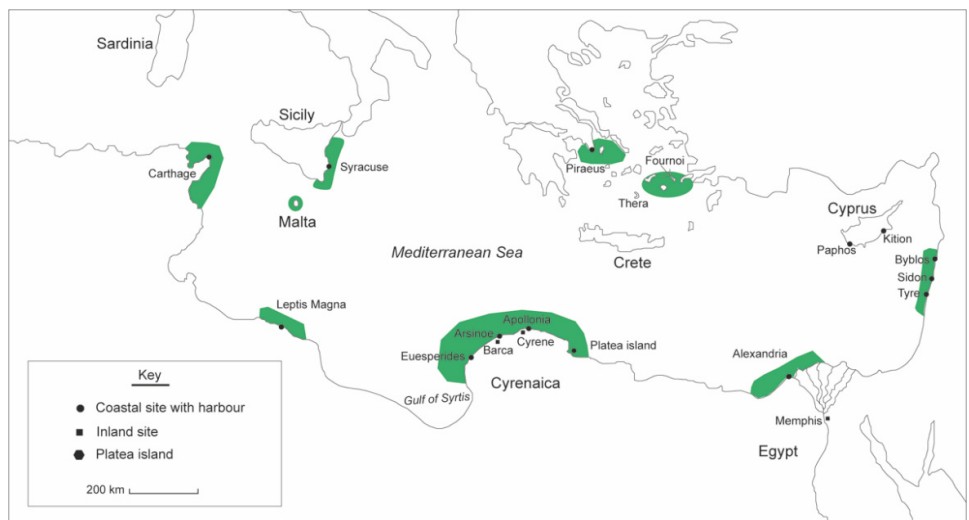

**Figure 7.** Map detailing areas where it may be possible to recover botanical remains of silphium from underwater archaeological sites.

A wide variety of organic remains have been recovered from Mediterranean shipwrecks. When a ship sinks into soft, anoxic sea mud after wreckage, the organic cargo items onboard can be preserved to a far greater extent than is commonly seen at terrestrial archaeological sites. Waterlogged botanical remains can thus offer greater insight into the diversity of plant use than carbonised remains simply because they occur in higher numbers [28] and are not dependent on proximity to fire to enter the archaeological record. Organic remains from ancient Mediterranean shipwrecks have included pomegranates [60], whole olives [26], almonds [61], grape seeds, pine pitch, butchered beef [31], and butchered pork [62]. As these trade goods have been preserved for thousands of years and recovered by underwater archaeologists, could this also be the case for silphium?

We have clear references to the maritime trade in silphium from contemporary primary sources: Theophrastus described how silphium was placed in amphorae 'when they are conveying it to Piraeus' [16]. As Apollonia was the principal harbour of the city of Cyrene, the best chance for the recovery of silphium or silphium-derived products is in and around the harbour at Apollonia. Underwater investigations at the harbour of Apollonia were conducted in the 1950s by a team led by Nicholas Flemming [25]. His team identified an inner harbour area and retrieved numerous artefacts including stone anchors. This inner harbour area is protected by both natural islands offshore and man-made harbour structures that run between the islands. As a relatively calm and protected area with a muddy seafloor, the inner harbour area of Apollonia may have the optimal conditions for the preservation of organic remains. The report on Apollonia given by Flemming did not describe any organic remains recovered from the seafloor. This was likely since these investigations were primarily a survey with limited recovery of artefacts. It would take excavations of the inner harbour seabed to effectively search for silphium.

In addition to the harbour areas of Apollonia, it is also a possibility that silphium could be found in amphorae from contemporary harbour sites around the Mediterranean, including the harbours of Malta, which is relatively nearby. The Maltese archipelago preserves countless underwater archaeological sites, some of which have clear connections to trade routes in North Africa, such as the Iron Age Xlendi shipwreck [63]. The ancient harbours of Eusperides, Carthage, and Leptis Magna could also contain remains of silphium due to their maritime connections to Apollonia and their geographical proximity to where silphium was produced. As Pireaus was specifically mentioned by Theophrastus as a destination for silphium exports, the sea around this harbour could contain the remains of shipwrecks with silphium onboard. Another interesting area that warrants further investigation is the veritable shipwreck graveyard of Fournoi. This Greek island features a combination of changeable currents, a rocky coastline, and a location along vital trade routes. Recent underwater investigations revealed the remains of multiple shipwrecks dating from the classical period to the medieval era [64]. If direct evidence for silphium can be found here, it may be possible to extract, sequence, and analyse the genetic makeup of this enigmatic plant.

There is tremendous potential for the recovery of ancient DNA from organic remains found at underwater archaeological sites [27]. This has been demonstrated recently by the successful recovery of well-preserved grape (*Vitis vinifera*) DNA from ancient, waterlogged grapes [65–67]. If silphium is recovered from underwater archaeological sites, it may be possible to extract ancient DNA from these remains. In addition, bioinformatic studies of the living candidate plants could be undertaken to scan for genetic varieties and pinpoint such alleles that are most typical to the Libyan region for each plant. Candidate plant genomes could easily be converted into a standard reference bioinformatic scan.

## 6. Conclusions

The silphium plant, also known as silphion, or Laserpicium, may be the most enigmatic plant and plant-based product of the ancient world. As the first recorded species extinction in world history [4], the nature of silphium, and the possible reasons for its extinction, warrant further research. As climate change increasingly threatens more terrestrial plant species with extinction [44], examples from the past may help us to better understand the mechanisms behind plant extinction and how to prevent it in the future.

The association between the silphium plant and contraception and abortifacients may explain its mysterious and elusive nature. Attitudes towards female chastity shifted in the 1st century CE when Augustus, the first Emperor of Rome, instituted his so-called 'law of the three sons' which encouraged marriage and population growth, while simultaneously barring young widows and celibate women of child-bearing age from inheriting property or attending public games, and making adultery both a private and public crime [68]. It is perhaps entirely coincidental that a plant product that was widely associated with inhibiting conception went extinct a mere 50 years after a law was passed increasing the punishment for extra-marital sex. What may be more likely is that the shift in attitude and perception towards extra-marital sex caused a decline in the demand for silphium and its unique properties. If the seasoning provided by silphium could be achieved by substituting asafoetida, with none of the contraceptive benefits, perhaps this substitution was encouraged.

The passage from Pausanias that makes oblique references to its properties as a contraceptive and abortifacient is especially illuminating [35]. If your maiden daughter went missing after two young gentlemen had been guests in your home, and you found silphium 'upon the table', this may have been the classical era equivalent of finding a soiled prophylactic on the floor. Our references to these properties are tantalisingly few, and perhaps this is the very reason why it is so difficult to grasp a clear picture of silphium. If its use for contraception was viewed as taboo in the ancient Greek world, then its use as an abortifacient was inevitably more so.

How and why this plant went extinct remain unclear. Indeed, recent scholarship questioned whether silphium survived in Anatolia [40], and therefore, would be classed as an extirpation from North Africa, rather than an extinction event. Complicating the picture is the fact that primary sources continued to make references to silphium for centuries after its supposed extinction: even in the 5th century CE, Synesius of Cyrene included silphium in his description of a contemporary cargo being shipped [10]. In this article, we have proposed several candidate species that could either be related to silphium, or silphium itself, but hiding in plain sight.

Cyrenaica, and Apollonia in particular, may yet be the sites of discovery of physical evidence of silphium. Carbonised botanical remains may be preserved, but we predict that underwater archaeological investigations will provide the first physical evidence of silphium, either in the form of waterlogged botanical remains of the actual plant, its fruit, or seeds, or perhaps more likely, silphium gum-resin inside a shipwreck amphora. Resins were an important class of agricultural product in the ancient world, and the trade in resins and their myriad uses—including as food, cosmetic ingredients, and incense—remain understudied. Far too little work has been conducted on the archaeological remains underwater off the coast of Libya, and coastal sites in this area are under continuous threat from wave action and coastal erosion [69]. While the early surveys conducted by Flemming and his team in the 1950s revealed a great deal of information about the harbour structures of ancient Apollonia [25], only a thorough investigation with excavation, artefact recovery, and scientific analysis of recovered artefacts is likely to reveal physical evidence of silphium. Future underwater investigations off the coast of Libya offer an exciting prospect for the recovery of this enigmatic plant.

**Author Contributions:** Conceptualization, L.B. and J.J.; methodology, L.B. and J.J.; software, L.B.; validation, L.B. and J.J.; formal analysis, L.B. and J.J.; investigation, L.B. and J.J.; resources, L.B. and J.J.; data curation, L.B. and J.J.; writing—original draft preparation, L.B. and J.J.; writing—review and editing, L.B.; visualization, L.B.; supervision, L.B. and J.J.; project administration, L.B. and J.J.; funding acquisition, N/A. All authors have read and agreed to the published version of the manuscript.

**Funding:** This research received no external funding.

**Acknowledgments:** We would like to thank Torbjörn Tyler, Mark Passehl, and Renzo Lucherini for their valuable help.

**Conflicts of Interest:** The authors declare no conflict of interest.

## Appendix A

**Table A1.** Ancient authors describing silphium.

| Author | Personal Details | Relevant Information/Quotes |
|---|---|---|
| Herodotus | Greek, contemporary (with silphium) | *Histories*, 4.169, Description of Cyrenaica<br>*Histories*, 4:145–205, Political history |
| Hippocrates | Greek, contemporary | Book 7, p. 547, Refers to how silphium can only be grown in Libya |
| Aristophanes | Greek, contemporary | Makes reference to silphium in many plays including *Birds*, *Knights*, *Plutus*. It was a fashionable luxury food item in Athens. Served with vinegar (acid) and salt. Possibly personal experience |
| Theophrastos | Greek, contemporary | Book 1, p. 165, Discovery of silphium<br>Book 2, pp. 13–21, Description T. uses the term 'Pherula-like' in a purely phenotypical way, for unrelated plants. Description: Leaves (*maspeton*) grow from the ground in spring. They fatten sheep, act as laxative for them. Later, stems emerge. The root should be cut, but not excessively |
| Strabo | Greek, contemporary | *Geography*, 2.2, 'These zones are remarkable for being extremely arid and sandy, and producing no vegetation with the exception of silphium' |
| Catullus | Roman, late contemporary | Poem 7 (about Lesbia), Refers to silphium-rich Libya in a love poem |
| Dioscorides | Greek, lived at time of extinction | *Materia Medica*, Book 3, The root is surrounded by a black membrane, and the juice went bad quickly, unless sifted with flour. Then it turned pale red and stayed good (as a resin?). No taste of garlic—unlike asafoetida |
| Pliny | Roman, lived at the time of extinction | Book 5, p. 547, Describes the inland fields of silphium |

| | | |
|---|---|---|
| Are-taeus | Greek, late Antiquity | Aret. *CA* 1.7 Chapter 6, Cure of Tetanus<br>'But if the stomach rejects this, give intermediately of the root of silphium an equal dose to the castor, or of myrrh the half of the silphium: all these things are to be drunk with honeyed water. But if there be a good supply of the juice of the silphium from Cyrene, wrap it, to the amount of a tare, in boiled honey, and give it to swallow.'<br>Aret. *CD* 1.2 Chapter 2, Cure for Cephalaea<br>'The diet in both kinds of the complaint should be light; little drink, water for drink, especially before giving any medicine, complete abstinence from acrid things, such as onions, garlic, the juice of silphium, but not altogether from mustard, for it acrimony, in addition to its being stomachic, is not unpleasant to the head…' *Medical uses* |
| Athe-naeus | Greek, late Antiquity | *The Deipnosophists* Ath. 14.17<br>'…the servant bastes the fish with vinegar: then there's Libyan silphium, dried in the genial rays of the midday sun.' *Description of a decadent banquet* |

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
