# Peer review of "Searching for Silphium: An Updated Review"

_heritage, doi:10.3390/heritage5020051_

Round 1

Reviewer 1 Report

This review article deals with an issue that has many intriguing controversies: the history of the mythical silphium plant, coveted throughout the ancient world, and mysteriously extinct in the 1st century AD.

Silphium was renowned for its medical properties and culinary value as a luxury spice, and was essential for the economy of ancient Cyrenaica (present day eastern Libya) where the plant was growing. Many classical authors mentioned it in plays or literary works, and several ancient coins and terracotta figurines from Cyrenaica depicted a silphium stem. Despite its great fame and use by most ancient Mediterranean cultures, Silphium inexplicably disappeared in the Roman period thus being the first recorded case of a plant extinction in world history. Many hypotheses have been formulated about the end of wild Silphium and also about the plant species identification, but none of them has been completely convincing so far.   

This review examines a great variety of information, from the historical background, classical sources, monetary and artistic iconography to medical properties and botanical interpretations. In addition, the authors have tried to investigate possible causes of silphium extinction formulating different reasonable explanations.

Overall, this paper provides a comprehensive overview of what is known about this plant. Moreover, waiting for new archaeobotanical and (ancient/seda)DNA data from underwater investigations at the main harbors along the trade routes of Cyrene, the authors propose a list of current plant species candidates for ancient Silphium, in an attempt to shed new light on the debated identification of this plant.

The manuscript has a lot of potential and could represent a good reference for the knowledge of this elusive plant so important for the ancient Mediterranean culture and economy. The paper has some (few) weaknesses/typos that should be corrected before being published:

- L 208-209 and 225-227: I would add the observation that the more or less realistic depiction of silphium depends also on the ‘phase’ of Roman art (realistic or idealistic/stylized) during which the coin or figurine was realized

- L 334: I would suggest changing ‘Avicenna ibn Sina’ to ‘Ibn Sina, known as Avicenna’ or ‘Avicenna (Ibn Sina)’

- L 375: Please, delete this title: it is the same of the subchapter 3.2. (L 343), and the related paragraph could be added to the section 3.2.

- L 460 and following: ‘Polyploidi’, ‘allopolyploidi’, ‘autopolyploidi’ = ‘Polyploidy’, ‘allopolyploidy’, ‘autopolyploidy’ and not in italics

- L 472: ‘heterosis or hybrid vigour’ = not in italics

- L 481: ‘sympodiums, apomictic reproduction’ = not in italics

- L 512: ‘Compositae’ is a family name and, according to botanical nomanclature, should not be written in italics. Please change also for Apiaceae, Umbelliferae, etc.

L 526: ‘Umbellifera’ = ‘Umbelliferae’

L 543: ‘Thapsia graganica’ = ‘Thapsia garganica’

L 551: ‘Thapsia garganica’ should be written in italics

L 628: Change ‘Lepcis Magna’ to ‘Leptis Magna’

- Figures 1 and 8: ‘Lepcis Magna’ = ‘Leptis Magna’

Author Response

Thank you for your comments on our paper. Our responses are marked below in bold.

L 208-209 and 225-227: I would add the observation that the more or less realistic depiction of silphium depends also on the ‘phase’ of Roman art (realistic or idealistic/stylized) during which the coin or figurine was realized

We have added this observation at line 233

L 334: I would suggest changing ‘Avicenna ibn Sina’ to ‘Ibn Sina, known as Avicenna’ or ‘Avicenna (Ibn Sina)’ We have made this change

- L 375: Please, delete this title: it is the same of the subchapter 3.2. (L 343), and the related paragraph could be added to the section 3.2. This has been deleted and the paragraph incorporated into the previous section, as suggested.

- L 460 and following: ‘Polyploidi’, ‘allopolyploidi’, ‘autopolyploidi’ = ‘Polyploidy’, ‘allopolyploidy’, ‘autopolyploidy’ and not in italics We have changed the spelling and removed the italics

- L 472: ‘heterosis or hybrid vigour’ = not in italics We have removed the italics from these terms

- L 481: ‘sympodiums, apomictic reproduction’ = not in italics We have removed the italics from these terms

- L 512: ‘Compositae’ is a family name and, according to botanical nomanclature, should not be written in italics. Please change also for Apiaceae, Umbelliferae, etc. We have removed the italics from these terms

L 526: ‘Umbellifera’ = ‘Umbelliferae’ We have changed this spelling

L 543: ‘Thapsia graganica’ = ‘Thapsia garganica’ We have fixed this type

L 551: ‘Thapsia garganica’ should be written in italics We have made this change as suggested

L 628: Change ‘Lepcis Magna’ to ‘Leptis Magna’ We have made this spelling change as suggested

- Figures 1 and 8: ‘Lepcis Magna’ = ‘Leptis Magna’ We have changed the spelling from Lepcis to Leptis in these Figures and re-inserted them into the document.

Reviewer 2 Report

This is an excellent paper on a very interesting topic. I only have two minor suggestions:

Line 187. Briggs is of course a good citation, but I would add Stefanie Jacomet, who spent her life working on waterlogged plant remains and has many important papers on this issue: Jacomet, S., 2013. Archaeobotany: Analyses of Plant Remains from Waterlogged Archaeological Sites, in: Menotti, F., O'Sullivan, A. (Eds.), The Oxford Handbook of Wetland Archaeology. Oxford University Press, Oxford, pp. 497-514.

Resins are rarely studied archaeologically. Maybe that is a point you could raise in the paper.

Author Response

Thank you for your review of our paper. I have consulted the Jacomet, S., (2013) publication and have incorporated information from this work into the manuscript at line 199 and line 615.

We have added a raised the point about resins being understudied in the Conclusion at line 704.